# Biological Activity of Propolis Ointment with the Addition of 1% Nanosilver in the Treatment of Experimentally-Evoked Burn Wounds

**DOI:** 10.3390/polym13142312

**Published:** 2021-07-14

**Authors:** Jakub Staniczek, Żaneta Jastrzębska-Stojko, Rafał Stojko

**Affiliations:** 1Department of Gynecology, Obstetrics and Gynecologic Oncology, Medical University of Silesia, Markiefki 87, 40-211 Katowice, Poland; rstojko@sum.edu.pl; 2Department of Anesthesiology, Central Clinical Hospital, Medical University of Silesia in Katowice, Medyków 14, 40-752 Katowice, Poland; zak@czkstojko.pl

**Keywords:** propolis, burns, sulfadiazine, apitherapy, nanosilver

## Abstract

The main objective of this study was to assess the pharmacological efficacy of ointments containing 1% propolis and 1% nanosilver, compared to the conventional treatment of burn wounds. In the evaluation of the results, we used clinical observation of scars, microbiological examinations, pathomorphological examinations, and analysis of free radicals. The analysis of the experiment results concerning the therapeutic effectiveness of the propolis ointment revealed its wide-ranging antibacterial action (against Gram-positive and Gram-negative bacteria). The 1% propolis ointment was found to accelerate neoangiogenesis and epithelialization, have a positive effect on the healing of burn wounds, improve the cosmetic look of scars, and have no side-effects. The analysis of free radicals in burn wounds showed impressive activity of the 1% nanosilver ointment in the reduction of free radicals. No synergism of pharmacological activity of propolis and nanosilver was shown. A comparative evaluation of the acquired research material allows us to provide a favorable opinion on the topical treatment of burn wounds with 1% propolis. The obtained results show that the 1% propolis ointment reduces healing time, offers antimicrobial action, and has a positive effect on the normal process of scar formation.

## 1. Introduction

A burn is injury of the skin, caused by thermal, chemical, or electric factors. Treatment of burn wounds is a significant problem in modern medicine. Every year, over 450,000 serious burn injuries that require medical treatment occur in the United States. One civilian fire death occurs every 2 h and 41 min. Scalding is the most common burn injury in children under four years old, accounting for 200,000 injuries per year. An estimated 50 percent of scalds are from spilled food and drinks. Each year, roughly 250,000 children under the age of 17 require medical attention for burn injuries. Roughly 15,000 children require hospitalization for burn injuries and about 1100 children per year die from fires and burn injuries. According to a 2016 report from the Agency for Healthcare Research and Quality, the total cost for the treatment of burns in 2010 was $1.5 billion, with another $5 billion in costs associated with lost work. Most of these injuries were preventable. An ideal topical preparation that fulfills the functions necessary for the effective treatment of thermal injuries is constantly in demand. Among the expected therapeutic properties are antimicrobial, regenerative, reparative, and anesthetic effects.

Currently, the most widely used topical treatment of burns is 1% silver sulfadiazine. The 1% silver salt of sulfathiazole, which is characterized by a synergistic effect of silver and sulfonamide, has been the standard in the local treatment of burns for several decades. The mechanism of the antimicrobial action of silver sulfadiazine (SSD) is based on the disruption of metabolic pathways essential for bacterial cell division and survival. The SSD-sensitive pathogens are *Pseudomonas aeruginosa*, *Klebsiella* species, *Enterobacter cloacae*, *Escherichia coli*, *Proteus mirabilis*, *Proteus vulgaris*, *Pseudomonas fluorescens*, *Serratia*, *Citrobacter*, *Staphylococcus aureus*, *Staphylococcus epidermis*, *Cyphylococcus aureus*, *Cyphylococcus epidermingidis*, and Enterobacter; and SSD shows in vitro activity against *Herpes virus* and *Candida albicans* [1,2,3].

However, there is an ongoing search for other substances of natural origin that can demonstrate better efficacy in the treatment of burn wounds. Attempts to use bee products for the treatment of thermal injuries have had promising results. Propolis has been proven to be very effective in the treatment of burn wounds. Propolis is used by bees to seal their nests. It dissolves easily in ethyl alcohol, thanks to which it is available in many forms, including globules, aerosols, ointments, or extracts. Due to its very high bactericidal, bacteriostatic, regenerative, and anesthetic activity, propolis has been used in numerous fields of medicine [4,5]. Propolis has a broad bacteriostatic and bactericidal effect. In addition, it prevents the development of mold, fungi, viruses, and protozoa [4,5,6].

The chemical composition of propolis depends on the local vegetation. The factors that inhibit the growth of bacteria in propolis are flavonoids, including pinocembrin, galangin, pinobanksin, and pinobanksin-3-acetate. Antibacterial activity is also demonstrated by aromatic esters, including phenylethyl and pentenyl esters of caffeic acid—caffeates, benzyl, and benzyl p-coumaric acid. Other flavonoids are less active: kaempferol, sakuranetin, isalpinine, pinostrobin, chrysin, tectochrizine, apigenin and pinostrobin chalcone, and hydroxy acids, including: caffeic, cinnamic, benzoic and ferulic acids. Other biotically active substances are aromatic acids: caffeic and a derivative of cinnamic acid, benzoic acid; hydroxy- or 1-methoxybenzoic acid and l-acetoxybetulenic sesquiterpene [4,5,6,7,8].

Antimicrobial activity is probably the result of a synergistic effect of sesquiterpenes, aromatic acids and flavonoids, aromatic esters, and other compounds in trace amounts. The most active substances were combined in various proportions, and it was found that the antibacterial and antifungal effects of propolis are only the sum of the activities of all microbiologically active ingredients [6,7,8,9,10,11]. In addition to the above-mentioned substances, propolis also contains other substances with antimicrobial properties: acetin, ramocitrin, quercetin, genquanin and naringenin, aromatic esters, including cinnamic acid ethyl esters, benzoic acid phenylmethyl esters, phenolic acids, as well as terpenes, sterols, and unsaturated fatty acids [11,12,13].

Propolis also has a regenerative effect in relation to damaged tissues. This is due to the content of compounds that stimulate cellular and systemic metabolism through the activation of ATPase and NADPH-2 tetrazolium reductase. These increase the mitotic index, accelerate tissue regeneration, and shorten healing time. In addition, it contains B vitamins and microelements such as calcium, manganese, lead, sulfur, aluminum, copper, nickel, cobalt, silicon, barium, chromium, zinc, tin, and others [14,15]. The development of nanotechnology and research on nanoparticles, and in particular on the most famous nanoproduct—nanosilver—has demonstrated their excellent antibacterial and antiviral properties.

Nanosilver particles are smaller than 100 nm and contain 20–15,000 silver atoms. The action of nanosilver leads to the degradation of bacterial, viral, and fungal cells. This effect is similar to that shown by antibiotics in the absence of pathogens being able to develop resistance [16,17].

The activity of nanosilver has been studied on numerous prokaryotic organisms such as bacteria, fungi, and viruses. Due to its strong antibacterial effect, nanosilver, apart from medical applications such as healing wounds and burns or as a disinfectant, is also used as an element of fabrics, but also as an implant coating. Nanosilver particles and Ag^+^ ions excreted by them easily destroy compounds containing sulfur and phosphorus, such as DNA and proteins [18,19,20,21,22].

Nanosilver affects the growth, movement, and reproduction of bacterial cells, and also blocks the respiration process and metabolic reactions taking place in the cell. The mechanism of the bactericidal activity of nanosilver is based on the surface and oligodynamic activity. The surface effect is mainly based on the destructive effect of nano Ag^+^ ions on the cell wall as well as by opsonization of the bacterial cell, hindering cell movement and the exchange of genetic material. A bacterial cell, surrounded by the layer of nanosilver, loses the ability to exchange genetic material between the cells and also loses the ability to reproduce through cell division.

The penetration of Ag^+^ ions into the cell interior is responsible for the oligodynamic effect, and hence inactivation of bacteria through the permanent binding of silver nanoparticles to proteins containing -SH groups, leading to the blocking of metabolic reactions taking place in cells. Ag^+^ ions deactivate the catalytic activity of enzymes, reacting with -SH groups, denature proteins, and perforate the cell wall. Silver nanoparticles block the process of building new bacterial cell walls, which are mainly made of murein and cysteine. The scheme of inactivation of bacterial cells by nanosilver is based on the catalysis of the process of converting oxygen and molecular oxygen ions into atomic oxygen with the ability to sterilize. Preventing the formation of new cell walls and cell death by degrading the existing cell wall occurs through the reaction of oxygen with cysteine thiol groups, leading to the formation of sulfide bonds between the amino acids. Nanosilver coating also destroys the cell membrane, affecting its potential, in such a way that it disrupts the sodium-potassium pumps, which in turn leads to cell swelling and disrupts the transport of sugars and amino acids to its interior. The catalytic properties of nanosilver lead to protein denaturation by creating free protons in bacterial cells, which break the disulfide bonds. Disorders in the respiration process of a bacterial cell consist of disrupting the flow of electrons in the cell, and this leads to a complete obstruction of the pathogen’s respiration. The mechanism of influence of nanosilver on fungi and viruses is analogous to the one described above for the bacterial cells. Nanosilver disturbs the water balance of fungi and affects the catalytic decomposition of the lipid-protein substrate of viruses. It is also believed that silver nanoparticles exhibit anti-inflammatory properties [23,24,25]. Reports on the harmfulness of nanosilver are ambiguous and are caused by the influence of many factors, such as the size, state of aggregation, or water solubility of nanoparticles. Experimental studies on animals show that there is a possibility of their absorption through the skin. Nanoparticles can reach individual organs (even the brain) and cause neurotoxic effects. Nanosilver has been shown to be genotoxic. It should be noted that the cytostatic doses exceed the antibacterial concentrations of nanosilver.

Taking into account the biotic activity of propolis and nanosilver, an experiment was designed based on a model experimental pattern of a burn wound, the assumption of which was to compare the effects of both active substances in the treatment of thermal damage and to verify the therapeutic effects with the commonly used drug, sulfathiazole, in the treatment of burns. In addition, it was planned to assess the synergism of action of nanosilver and propolis by making an ointment with a mixture of 1% propolis and 1% nanosilver.

## 2. Experimental Section

The experiment concerned the observation of burn wound healing. The experiment was approved by the Local Ethics Committee for Animal Experiments in Katowice (Resolution No. 66/2014 of 30 July 2014). Burn wounds were made according to the standard Hoekstra model, in accordance with the Dutch Animal Testing Law and the current experimental protocol of the Charity Committee at the University of Amsterdam [26].

The experiment lasted for 25 days and was divided into six control days (0, 5, 10, 15, 20, 25), on which bacteriological swabs were taken from the wounds and surgical sections were made for further analysis. On each day of the experiment, the burn wounds were applied twice with ointments, according to a group distribution scheme.

The ointment recipes were developed at the Polish Apitherapy Foundation in Katowice and then made at the Department of Hygiene, Bioanalysis and Environmental Research, the Faculty of Pharmacy in Sosnowiec, the Medical University of Silesia in Katowice.

The experiment used two Polish Large White (PLW) pigs 16 weeks of age, weighing about 40 kg, from the breeding of the Institute of Animal Production in Grodziec Śląski. During the experiment, full psychophysical comfort was maintained in the animals, which did not affect breeding well-being due to the therapy used. The animals were kept under the same zoohygienic conditions and were fed the same wholesome feed. Prior to the start of the experiment, the animals underwent a two-week breeding quarantine period. Invasive procedures were performed under general anesthesia. Premedication, analgesia, and anesthesia were managed according to the standard recommendations and an established dosage schedule.

On day 0, the animals were premedicated with atropine sulfate, 0.05 mg/kg, ketamine hydrochloride, 3 mg/kg, and xylazine hydrochloride, 1 mg/kg.

Then, for analgesia and anesthesia, thiopental was administered into the marginal vein of the ear at a dose of 5 mg/kg, fentanyl at a dose of 0.5 µg/kg, followed by making 18 identical burn wounds in each of the experimental animals; symmetrically distributed, 9 on both sides along the axis of the spine. A 3 cm × 1.5 cm brass block with a weight of 10 dag, heated up to 170 °C, was applied to the skin surface for 10 s. The total area of the burn wounds covered 90 cm^2^, i.e., about 2% of the body surface of the animal subjected to the experiment. During the procedure, vital signs such as ECG and saturation were monitored and, if necessary, respiratory support was used using a respirator.

After making 36 burn wounds, the animals were divided into 2 groups: D—experimental group (pig I right and left side and pig II right side), and K—control group (pig II left side). Then, group D was divided into subgroups: D1—left side of pig I (9 burn wounds); D2—right side of pig I (9 burn wounds), and D3—right side of pig II (9 burn wounds).

Wounds in the D1 group were treated with a 1% propolis ointment every 12 h for 25 days, wounds in the D2 group with a 1% nanosilver ointment with the same frequency, and wounds in the D3 group were treated with an ointment with 1% propolis and 1% nanosilver with the same frequency.

The subgroup K consisted of 9 burn wounds on the left side of pig II, to which sulfathiazole silver salt was applied every 12 h.

Tissue sampling assumed surgical excision of skin blocks approximately 20 mm × 5 mm in size, which contained skin from the center of the burn wound, the wound part, and a fragment of healthy skin. For each group, 2 sections were made on the control day (one for histopathological examination and one for biophysical examination). A total of 25 skin blocks were collected for histopathological examination, 6 for each of the experimental and control groups. In addition, a fragment of healthy skin was also collected for comparison purposes. The skin samples were fixed in formaldehyde.

## 3. Results

### 3.1. Clinical Assessment

During the 25-day observation period of burn wounds, dynamically occurring changes in the healing process were observed in the following days of the experiment, depending on the therapeutic agents used.

In the group treated with the 1% propolis ointment (D1), the wound treated with this formulation showed a significant progress in regenerative processes since the first days of the experiment. The wound surface gradually reduced from day 10 to a size of about 1 cm × 1 cm on day 25 of the experiment. Swelling and redness of the tissues surrounding the burn showed regression from day 10 of the experiment, up to a complete disappearance of inflammation on day 15. Burn wounds from each group were also constantly monitored beyond the control days, and so, in group D1 treated with the 1% propolis ointment, rapid scab wound covering was found as early as day 7 of the experiment and complete wound epithelization was observed as early as on day 13.

Group D2 treated with the 1% nanosilver ointment showed a slightly similar clinical picture to the group D1; however, there was a delay of about 2–3 days in relation to the t1% propolis ointment—a scab covered the wound approximately 10 days after the experiment and complete epithelization was observed approximately 15 days from the experiment. As gradual contraction of the wound followed later, from day 15 of the experiment, the size of the wound reached 1 cm × 1 cm on day 25.

Group D3 treated with the ointment with 1% propolis and 1% nanosilver showed a morphological difference in the early days of the experiment. Until day 10 of the experiment, the wound was partly covered with a dry scab, strongly adhering to the wound. The burn surface in group D3 systematically reduced since day 20, reaching a dimension of about 1.5 cm × 1.5 cm on the final day of the experiment. There was no synergy of active substances, so the course of therapy and the effect can be based on the activity of the propolis. During the clinical observations, the wound was also covered with a crust on day 15, and full epithelization occurred on day 20 of the experiment.

In group K, treated with sulfathiazole, the wound area slightly decreased until the end of the experiment and it reached a size of 2 cm × 2 cm. On day 25 of the experiment, the wound was still covered with a hard and dry scab, which bled intensively on attempts to remove it. Intense granulation was visible under the scab. On day 25, the wound still had no epithelization features.

Analyzing the clinical observations of the individual groups, it should be stated that in the D1 group treated with the 1% propolis ointment, scab formation, epithelization, as well as the cosmetic effect stood out from the rest of the experimental groups. Slightly worse effects in comparison with the 1% propolis ointment in clinical advancement were found for the 1% nanosilver ointment (group D2). The cosmetic effect in this group showed a significant similarity to the propolis treatment; however, the individual intermediate stages of scar formation showed a delay in occurrence of several days.

It is worth noting that in the D3 group treated with the ointment with 1% propolis and 1% nanosilver, no advantage in therapy was demonstrated. There was a lack of synergism of action of both active substances, and even blocking of the individual action of propolis and nanosilver was found.

Silver sulfathiazole (Sulfathiazole) salt therapy did not show significant progression since the first days of the experiment. Throughout the experiment, the wound slightly shrank and was covered by a hard, cracked scab, strongly adhering to the wound surface, which bled profusely on attemps to pull at it. Granulation tissue was present under the crust until day 25.

### 3.2. Microbiological Tests

The aim of the study was to perform a semi-quantitative and qualitative bacteriological analysis of microorganisms cultured from the swabs taken from the wounds on day zero and remaining in them or appearing de novo during treatment, on subsequent days, as a consequence of the local therapy. The obtained bacteriological results were compared with the microbiological results from the swabs taken at the same time intervals, from two different, independent places of the skin of a healthy animal. The methodology of the semi-quantitative study was to determine the degree of bacterial growth: (+++) very abundant growth, (++) abundant growth, (+) scanty growth. The last stage of the microbiological test was to identify the species of bacteria grown and determine the drug resistance. During the experiment, 36 cultures were obtained from the burn wounds.

#### 3.2.1. Qualitative and Semi-Quantitative Analysis

##### Healthy Skin

In the course of the experiment, pathogens that are typical for the physiological flora of the skin were identified in swabs taken from the healthy skin. They were dominated by G(+) bacteria, G(−) bacteria and fungi. In the experiment, Staphylococcus epidermidis MSSE, *Staphylococcus aureus* MSSA, *Streptococcus* B-haemoliticus and G(−) bacilli were identified, such as *E. coli*, *Pseudomonas aeruginosa* or *Klebsiella pneumoniae*. The exact division of healthy skin pathogens, divided into individual days of the experiment, is shown in Table 1 and Table 2.

##### Wounds Treated with the 1% Propolis Ointment

In the wounds treated with the 1% propolis ointment, during the study compared to healthy skin, a significant decrease in the number of G(+) colonies was observed, up to their complete absence in the culture results, on days 20 and 25 of the experiment. On 15 day of experiment, two colonies of *Escherichia coli* were observed (Table 3).

The number of colonies of two strains—*Escherichia coli* and *Proteus mirabilis*—remained variable, but with a downward trend, and a significant increase was observed in the number of *Pseudomonas aeruginosa* and *Klebsiella pneumoniae* colonies, which dominated in cultures obtained on days 20 and 25 of the study.

The increasing amount of G(−) bacteria is a consequence of the elimination of G(+) bacteria, against which propolis has a proven and indisputable bactericidal effect. It also has a similar activity against G(−) rods. The exact division of pathogens grown from the wound treated with 1% propolis (group D1), broken down into individual days of the experiment, is presented in Table 3.

##### Wound Treated with the 1% Nanosilver Ointment

In wounds treated with the 1% nanosilver ointment, microorganisms were isolated from the collected material and culture was carried out, whose composition resembled the microorganisms present in cultures taken from healthy skin. This indicates a total lack of interaction of the agent used with the physiological flora of the skin, and bacteria appear in the wound, spreading through the continuity of the skin adjacent to the wound.

The exact division of pathogens grown from the wound treated with the 1% nanosilver ointment, divided into individual days of the experiment, is presented in Table 4.

##### Wounds Treated with the Ointment with 1% Propolis and 1% Nanosilver

On 5. day of experiment, two colonies of *Escherichia coli* were observed (Table 5).

In the cultures collected on subsequent days of the experiment, from wounds treated with 1% propolis and 1% nanosilver, a gradual decrease up to complete elimination of G(+) bacteria was observed—the same was noted on days 20 and 25 of the experiment for G(−) bacteria *Pseudomonas aeruginosa* or *Proteus mirabilis*.

These results confirm the weak antibacterial effect of NanoAg and indicate that the active substance in the mixture is propolis. The composition obtained and trends in the number of colonies of individually identified bacteria are comparable with the results obtained from the inoculation of material taken from wounds treated with the 1% propolis ointment.

The exact division of pathogens cultured from the wound treated with the ointment with 1% propolis and 1% nanosilver, broken down into individual days of the experiment, is presented in Table 5.

##### Wounds Treated with Silver Salt of Sulfathiazole

In wounds treated with silver salt of sulfathiazole, compared to healthy skin, a downward trend was observed in the number of Staphylococcus epidermidis MSSE colonies; however, *Staphylococcus aureus* MSSA and G(+) *Escherichia coli* and *Proteus mirabilis* remained.

The results demonstrate that in the conducted study, the silver salt of sulfathiazole had little effect on the selected bacteria, both G(+) and G(−). The exact division of pathogens grown from the wound treated with the silver salt of sulfathiazole (group K), divided into individual days of the experiment, is presented in Table 6.

### 3.3. Drug Resistance Analysis

#### 3.3.1. Staphylococcus Epidermidis MSSE

The experiment also evaluated the drug susceptibility of the cultured bacterial strains to commonly used antibiotics. It was observed that Staphylococcus epidermidis MSSE cultured from swabs taken from the healthy skin of pig I, on day 0 and on the following days of the experiment, showed sensitivity to each of the antibiotics used in the antibiogram. The results from healthy skin of pig II were different—MSSE showed resistance to ciprofloxacin, clindamycin, and trimethoprim/sulfamethoxazole on day 0 until the end of the study. Such a different result may be due to a different bacterial flora of pig II. In the case of wounds treated with the 1% propolis ointment, Staphylococcus epidermidis MSSE cultured during the experiment showed resistance to erythromycin and clindamycin on 0 to the last day of the experiment. However, it remained sensitive to trimethoprim/ sulfamethoxazole. In contrast, in wounds treated with the 1% propolis ointment and 1% nanosilver ointment, and ulfathiazole, MSSE resistance is identical to the resistance pattern obtained for the healthy skin of pig II. Staphylococcal resistance is even greater in wounds treated with the 1% nanosilver ointment and also includes amikacin.

#### 3.3.2. *Staphylococcus aureus* MSSA

*Staphylococcus aureus* MSSA strains grown from swabs taken from healthy pigs I and II showed resistance to ciprofloxacin, clindamycin, erythromycin, and trimethoprim/sulfamethoxazole. These are not common strains sensitive to commonly used antibiotics.

The resistance profile of *Staphylococcus aureus* MSSA grown from the wounds treated with the 1% propolis ointment, the 1% nanosilver ointment, and the ointment with a mixture of 1% propolis and 1% nanosilver was identical. Sensitivity to aminoglycosides and cloxacillin (semi-synthetic penicillin) was maintained. The resistance profile of *Staphylococcus aureus* MSSA slightly changed in sulfathiazole-treated wounds. During the treatment, on day 25, *Staphylococcus aureus* MSSA was resistant to cloxacillin, an antibiotic that is very effective against G(+) granulomas.

#### 3.3.3. *Streptococcus*
*β**-haemolyticus* Group C

*Streptococcus β**-haemolyticus* group C, bred both from the cultures of healthy skin of pigs I and II, and at each stage of the study, in the cultures taken from the wounds treated with the individual test preparations, was characterized by resistance to erythromycin and clindamycin. However, sensitivity to ampicillin remained.

#### 3.3.4. *Escherichia coli*

*Escherichia coli* grown in cultures taken from the healthy skin of pigs I and II was characterized by resistance to amoxicillin/clavulanic acid and trimethoprim/sulfamethoxazole.

In wounds treated with the 1% propolis ointment and the 1% nanosilver ointment, the identified *E. coli* showed primary resistance to amoxicillin with clavulanic acid, while the initial sensitivity to trimethoprim/sulfamethoxazole changed in the subsequent cultures on days 15 and 2 to resistance. Sensitivity to aminoglycosides, fluoroquinolones, and second generation cephalosporins was maintained. In wounds treated with a mixture of 1% propolis and 1% nanosilver and sulfathiazole from days 0 to 25, resistance to these antibiotics was maintained.

### 3.4. Histopathological Findings

At the Department of Pathology of the Medical Faculty in Katowice, the Medical University of Silesia in Katowice, after prior dehydration of the sections with ethanol, acetone, and xylene, paraffin blocks were prepared, and then cut into 7-micron thick layers stained with hematoxylin and eosin. The preparations prepared in this way were viewed under a microscope at 40×, 100×, 200× and 400× magnification.

The histopathological observation of preparations taken from burn wounds, in the following stages of healing, proceeded as follows: on day 0 of the experiment, no differences between the experimental and control groups were observed. The wounds were characterized by the lack of inflammatory infiltration, lack of epidermization and neovascularization. Superficial necrosis was present in all the wounds.

On day 5 of the experiment, slight differences in the wound healing process were observed in the microscopic image. Inflammation occurred in all wounds and the surrounding tissues, however with varying severity, type, and depth of infiltration. In the D1 group, treated with the 1% propolis ointment, there was a medium dense granulocytic infiltration reaching the dermis. The wounds in the D2 group, treated with the 1% nanosilver ointment, were characterized by the granulocytic infiltration of low density and the depth of infiltration reaching the dermis. In the wounds treated with a mixture of 1% propolis and 1% nanosilver, there was a large inflammatory granulocytic infiltration, reaching as far as the fat tissue. Sulfathiazole-treated wounds were characterized by a small, mixed infiltration, reaching the dermis. On day 5 of the therapy, all wounds from the experimental and control groups did not show epidermization. Superficial necrosis was low in the experimental groups and significant necrosis was observed in the control group treated with sulfathiazole. Clinical observations from day 5 are presented in Figure 1, Figure 2, Figure 3 and Figure 4.

On day 10, all wounds in the control and experimental groups were characterized by a moderate inflammatory infiltration, with a predominance of granulocytes reaching the dermis. In the D1 experimental group, inflammation occurred at the same level as on the 5th control day. Experimental groups D2 and control K, treated successively with the 1% nanosilver ointment and sulfathiazole, were characterized by a greater intensity of inflammation, compared to the fifth control day. In group D3, treated with a mixture of 1% propolis and 1% nanosilver, a regression of inflammatory infiltration, from high to medium and depth, from adipose tissue to the dermis, was noted. In the experimental group D1, treated with the 1% propolis ointment, and control K, treated with sulfathiazole, the appearance of slight epidermization was seen on day 10. In addition, on day 10, all wounds from the experimental and control groups showed slight superficial necrosis and were characterized by a moderate neovascularization.

On day 15, the wound sections examined varied considerably, depending on the therapeutic agent used. The wound treated with the 1% propolis ointment was constantly characterized by slight epidermization and medium neovascularization, as was on day 10; however, the wound did not show superficial necrosis. In addition, the type of inflammatory infiltration changed to lymphocytic-granulocytic. The wound from the experimental group D2, treated with the 1% nanosilver ointment and the wound from the control group K, treated with sulfathiazole, constantly showed progression of inflammation, from medium to large. Wounds from the experimental group D2 and control K, on day 15 were covered with large superficial necrosis. The wound treated with the 1% nanosilver ointment showed high neovascularization with no epidermization. The wound from group D3, treated with a mixture of 1% propolis and 1% nanosilver, showed constant inflammatory infiltration of a medium degree, however, with a change in type on day 1 to lymphocytic-granulocytic. In addition, on day 15, the wound from the experimental group D3 was covered with small superficial necrosis, with medium-grade neovascularization. The wound showed no signs of epidermization. Clinical observations from day 15 are presented in Figure 5, Figure 6, Figure 7 and Figure 8.

On day 20, the wound surface in group D1 treated with the 1% propolis ointment and in group D2 treated with the 1% nanosilver ointment was shown to decrease. The wounds were scarred in both groups with areas of visible epithelialization. Additionally, in group D1, the wound was covered with a small amount of soft scab. In both groups, the area of the tissues surrounding the burn did not show any signs of inflammation. In group D3, where the burn wound was treated with the ointment containing 1% propolis and 1% nanosilver, the wound was covered with a scar with numerous epithelial foci. The wound showed no signs of shrinkage and no inflammation in the wound area. In group K (treated with sulfathiazole), no significant healing features were observed. The wound was covered with a hard and dry scab. There was visible bleeding in the area where the scab was separated from the wound. Small clusters of granulation tissue were seen under the scab.

On day 25, in group D1 treated with the 1% propolis ointment, a significant reduction in wound size was observed, as compared to the previous control periods. The wound treated with the 1% propolis ointment showed complete healing. The wound was covered with a soft scar and the wound zone was not inflamed. The wound in group D2, treated with the 1% nanosilver ointment, was characterized by a reduction in area compared to the previous control days. The wound was characterized by a hard scar, partially covered by soft, tiny scabs. The tissues surrounding the wound showed no inflammation features. The wound in the control group D3, treated with the ointment with 1% propolis and 1% nanosilver, showed a hard scar with a few epithelial sites, after the previous scabs. The wound surface was visibly reduced, without any change in the image of the surrounding tissues.

In group K, treated with sulfathiazole, it was observed that the wound was partially covered with a hard, dry scab strongly adhering to its surface. The scab partially protruded at the edge of the wound; spot bleeding was visible at the site of the scab separation. The granulation tissue was visible under the scab. The clinical stage is presented in Figure 9, Figure 10, Figure 11 and Figure 12.

### 3.5. Histopathological Results

The histopathological observations of the preparations taken from burn wounds in the subsequent stages of healing were described below. On day 0 of the experiment, no differences were observed between the experimental and control groups. The wounds were characterized by lack of inflammatory infiltration, no epidermization, and no neovascularization. A superficial necrosis was present in all wounds. The histopathological findings from day 0 are presented in Table 7.

On day 5 of the experiment, slight differences in the wound healing process were observed. Inflammation was present in all wounds and the surrounding tissues, however, with a different intensity, type and depth of infiltration. In group D1, treated with the 1% propolis ointment, there was a moderately dense granulocytic infiltration reaching the dermis. Wounds in group D2, treated with the 1% nanosilver ointment, were characterized by a granulocytic infiltration with a low density and depth of infiltration reaching the dermis. In wounds treated with a mixture of 1% propolis and 1% nanosilver, there was a large granulocytic inflammatory infiltration reaching the depth of the adipose tissue. Wounds treated with sulfathiazole were characterized by a small, mixed infiltration reaching the dermis. On day 5 of the treatment, all wounds from the experimental and control groups did not show any epidermization features. Superficial necrosis was low in the experimental groups and a high degree of necrosis was observed in the sulfathiazole-treated control group. The histopathological findings from day 5 are presented in Table 8 and Figure 13 and Figure 14.

On day 10, all wounds in the control and experimental groups were characterized by a moderate inflammatory infiltration, with a predominance of granulocytes, reaching the dermis. In the experimental group D1, inflammation was at the same level as on the 5th control day. The experimental group D2 and the control group K, treated successively with the 1% nanosilver ointment and sulfathiazole, showed a greater intensity of inflammation in relation to the 5th control day. In group D3, treated with a mixture of 1% propolis and 1% nanosilver, a regression of the inflammatory infiltration, from large to medium, and depth, from adipose tissue to the dermis, was observed. In the experimental group D1, treated with the 1% propolis ointment and control K, treated with sulfathiazole, on day 10, a slight epidermization appeared. On day 10, additionally, all the wounds from the experimental and control groups showed slight superficial necrosis and were characterized by a moderate neovascularization. The histopathological findings from day 10 are presented in Table 9 and Figure 15, Figure 16 and Figure 17.

No epidermization, with slight necrosis (40×, H-E) in the wound treated with the ointment with 1% propolis and 1% nanosilver, on day 10.

On day 15, the wound samples examined differed significantly, depending on the therapeutic agent used. The wound treated with the 1% propolis ointment was characterized by a consistently slight epidermization and moderate neovascularization, as was on day 10; however, the wound showed no foci of superficial necrosis. Additionally, the type of inflammatory infiltration changed into a lymphocytic-granulocytic one. The wound in the experimental group D2, treated with the 1% nanosilver ointment, and the wound in control group K, treated with sulfathiazole, continuously showed the progression of inflammation, from moderate to severe. On day 15, the wounds from the experimental group D2 and control group K were covered with large superficial necrosis.

The wound treated with the 1% nanosilver ointment showed high neovascularization, with no epidermization at the same time. The wound from group D3, treated with a mixture of 1% propolis and 1% nanosilver, showed a constant moderate inflammatory infiltration, however, with a change of type on day 15 to lymphocytic-granulocytic. Additionally, on day 15, the wound in the D3 experimental group was covered with a slight superficial necrosis with moderate neovascularization. The wound showed no signs of epidermization. The histopathological findings from day 15 are presented in Table 10 and Figure 18 and Figure 19.

On day 20, the histopathological appearance varied significantly. In wound D1 treated with the 1% propolis ointment, the intensity and nature of the inflammation regressed. A small inflammatory lymphocytic infiltration was found. An increase in epidermization was also noted, from being slight on day 15 to high on day 20. Additionally, the wound was slightly neovascularized.

The wound from the experimental group D2, treated with the 1% propolis ointment, showed a regression of inflammation. The wound began to show a slight epidermization only on day 20. There was also a decrease in neovascularization, from high on day 15 to low on day 20, and no superficial necrosis was found. Wound D3, treated with a mixture of 1% propolis and 1% nanosilver, showed no changes on day 20 in the intensity and type of the inflammatory infiltration. Just like the wound in group D2, it was characterized by a slight epidermization only on day 20, with the simultaneous development of superficial necrosis, from being slight on day 15 to large on day 20. The wound was highly neovascularized. In the wound from the control group K, treated with sulfathiazole, the intensity and type of the inflammatory infiltration did not change on day 20, and additionally, epidermization was inhibited. The histopathological findings from day 20 are presented in Table 11 and Figure 20 and Figure 21.

On the last day of the follow-up, it was found that wounds D1 and D2 had similar features. They were characterized by a lack of inflammation and no superficial necrosis. The wounds showed high epidermization with a simultaneous slight neovascularization.

Wound D3, treated with a mixture of 1% propolis and 1% nanosilver, on day 25, consistently showed a moderate lymphocytic inflammatory infiltration reaching the dermis. On day 25, an increase in epidermization, from slight to high, and a reduction of superficial necrosis were observed. Neovascularization also regressed. Wounds from the control group K, treated with sulfathiazole, were constantly characterized on day 25 by superficial necrosis, slight epidermization, and regression of neovascularization. Burn wounds treated with sulfathiazole were characterized by the presence of a lymphocytic-granulocytic inflammatory infiltration, with low intensity, reaching the dermis. The histopathological findings from day 25 are presented in Table 12 and Figure 22 and Figure 23.

### 3.6. Analysis of Reactive Oxygen Species

The research on the amount of reactive oxygen species was carried out at the Department of Biophysics, Faculty of Pharmacy with the Division of Laboratory Medicine in Sosnowiec, Medical University of Silesia in Katowice.

#### 3.6.1. Preparation of Burn Wound Samples for EPR Spectra Measurements

Skin sections from the untreated burn wounds and wounds during treatment were placed in thin-walled glass tubes with an outer diameter of 3 mm. Empty measuring tubes did not produce EPR signals at the maximum line gains and maximum microwave power (70 mW) was used.

#### 3.6.2. Conditions for EPR Spectra Measurement

EPR spectra of the tested skin samples were recorded at room temperature using an X-band paramagnetic resonance electron spectrometer (9.3 GHz) with a magnetic field modulation of 100 kHz, by RADIOPAN (Poznań, Poland). The frequency of microwave radiation was recorded with the MCM 101 m, manufactured by EPRAD (Poznań, Poland). The maximum microwave power generated by the klystron was 70 mW. Skin samples in the measuring tubes were placed between the poles of the electromagnet and EPR spectra were computer-recorded in the form of the first derivative of absorption. A numerical data acquisition system and a software for spectroscopic analyses by JAGMAR (Krakow, Poland) with LabVIEW 8.5 software by NATIONAL INSTRUMENTS (1500 N Mopac Expwy, Austin, TX, USA) were used.

EPR spectra were measured using microwave power in the range of 2.2–70 mW. The power of microwave radiation used during the measurement of EPR spectra was calculated using the formula defining the system attenuation in decibels: Attenuation (dB) = 10 lg (Mo/M), where:Mo—the total power of microwave radiation produced by the klystron (70 mW),M—microwave radiation power used when measuring the EPR spectrum.

The maximum attenuation applied was 15 dB.

For the tested samples, the parameters of EPR spectra recorded at a low microwave power of 2.2 mW, which conditioned the lack of microwave signal saturation, were compared. The influence of microwave power (2.2–70 mW) on the shape and parameters of EPR spectra of individual samples was also investigated.

##### Analyzed Parameters of EPR Spectra

The following parameters of EPR spectra were determined: line amplitude (A (j. Resp.)), line width (ΔB_pp_ (mT), integral line intensity (I (j. Relative)) (Figure 24). The parameters of EPR spectra were determined using SWAMPv1.2 software from JAGMAR (Krakow, Poland).

The amplitude (A) of the EPR line increases with the rise in the number of free radicals present in the sample. The width (ΔB_pp_) of the EPR line depends on the magnetic interactions between the unpaired electrons in the sample. Dipole interactions widen the EPR lines, and exchanged interactions can cause a narrowing of the lines. The integral intensity (I) of the EPR line is the area under the resonance absorption curve. The integral intensity of the line written as the first derivative of absorption was determined by double integration. The integral intensity of the EPR line is used to calculate the concentration of free radicals in the sample. The concentration of (N) free radicals in the sample is proportional to the intensity of the integral EPR line.

#### 3.6.3. Determination of the Concentration of Free Radicals in the Samples

The concentration (N) of free radicals in the tested skin samples was determined according to the formula: N = n_u_ (I_p_A_ru_W_u_)/(I_u_A_rp_W_p_m). Ultramarine (sulfur-containing sodium aluminosilicate) was used as the concentration standard, which shows high stability of the paramagnetic centers present in it. In order to increase the accuracy of the measurement, the so-called internal standard—ruby crystal (Al_2_O_3_:Cr^3+^)—was inserted into the resonator below the tested sample using a goniometer.

#### 3.6.4. Determination of the Effect of Microwave Power on the Amplitude and Width of the EPR Line

The effect of microwave power on the amplitude (A) and width (ΔB_pp_) of the EPR lines of the burn wound samples was analyzed in order to determine the nature (homogeneous or heterogeneous) of spectral broadening. The amplitude of the extended EPR line uniformly increased with the increase in microwave power, and decreased on reaching the maximum value. The width of the widened line grew uniformly with the increase in microwave power.

The amplitude of the EPR line widened non-uniformly after reaching the maximum value and did not change with a further increase in microwave power (Figure 25).

The width of the widened line did not change uniformly with the microwave power (Figure 26)

#### 3.6.5. Analyzed Parameters of the Shape of EPR Spectra

The study analyzed the shape of EPR spectra in order to confirm the hypothesis on the complex system of free radicals in the tested samples of burn wounds. The presence of an asymmetric spectrum with asymmetry parameters dependent on the microwave power used during line measurements indicates the presence of several types of free radicals in the samples.

For spectra recorded in the microwave power range of 2.2–70 mW, the following parameters of EPR line asymmetry were determined: A1–A2, |A1–A2, B1–B2, |B1–B2|, and A1/A2, B1/B2. It was found that the changes in EPR line asymmetry with microwave power were best illustrated by the A1/A2 parameter. Therefore, the paper presents the results of the analyses of this parameter.

#### 3.6.6. Results of Research on Reactive Oxygen Species in Burn Wounds Using Electron Paramagnetic Resonance Spectroscopy

##### Measurement of Free Radicals in Healthy Skin

For the healthy skin sample, a slight asymmetric EPR spectrum was recorded (Figure 27). The EPR spectrum was acquired with a signal gain of 4 × 104 and a microwave power of 2.2 mW. A large asymmetric EPR spectrum was recorded for the initial burn wound sample on day 0, not treated with therapists (Figure 28). EPR spectra were acquired with a signal gain of 4 × 104 and microwave power of 2.2 mW. A large asymmetric EPR spectrum was recorded for the initial burn wound sample on day 0, not treated with therapists (Figure 28). EPR spectra were acquired with a signal gain of 4 × 104 and microwave power of 2.2 mW.

##### Effect of Treating Burn Wounds with the 1% Propolis Ointment on Free Radicals

EPR spectra were recorded for wounds on days 5, 10, 15, 20, and 25 of treatment with the 1% propolis ointment. EPR spectra of the tested wound samples are shown in Figure 29a–e, respectively. EPR spectra were recorded at room temperature with a low microwave power of 2.2 mW (15 dB attenuation). The spectra of all the tested samples are broad asymmetric lines.

##### The Effect of Treatment of Burn Wounds with the 1% Nanosilver Ointment on Free Radicals

EPR spectra were recorded for the wounds on days 5, 10, 15, 20, and 25 of the treatment, shown in Figure 30a–e, respectively. EPR spectra shown in Figure 30a–e were recorded at room temperature with a low microwave power of 2.2 mW (15 dB attenuation). The spectra of all the tested samples are broad asymmetric lines.

##### Effect of Treating Burn Wounds with the Ointment with 1% Propolis and 1% Nanosilver on Free Radicals

EPR spectra of the wounds were recorded on days 5, 10, 15, 20, and 25 of the treatment with the ointment with 1% propolis and 1% nanosilver. EPR spectra of the tested wound samples are shown in Figure 31a–e, respectively. EPR spectra were recorded at room temperature with a low microwave power of 2.2 mW (15 dB attenuation). The spectra of all the tested samples are broad asymmetric lines.

##### Effect of Sulfathiazole Treatment of Burn Wounds on Free Radicals

EPR spectra of the wounds were recorded on days 5, 10, 15, 20, and 25 of sulfathiazole treatment. EPR spectra of the tested wound samples are shown in Figure 32a–e, respectively. EPR spectra were recorded at room temperature with a low microwave power of 2.2 mW (15 dB attenuation). The spectra of all the tested samples are broad asymmetric lines.

#### 3.6.7. Analysis of Free Radical Concentration Measurement in the Experimental and Control Groups

The concentration of free radicals in the treated burn wounds ranged from 1.6 × 1018–19.5 × 1018 spin/g. A decrease in the concentration of free radicals was observed along with the duration of treatment with all therapists participating in the experiment.

The highest concentration value (N) of free radicals was observed for the burn wound treated with the ointment with 1% propolis and 1% nanosilver on day 10 of the therapy (19.5 × 1018 spin/g).

The lowest value of the concentration (N) of free radicals was observed for the burn wound treated with the 1% nanosilver ointment on day 25 of the therapy (1.6 × 1018 spin/g) (Figure 33).

On day 5 of the treatment, the lowest concentration of free radicals was seen in the wound treated with the 1% nanosilver (8.9 × 1018 spin/g) ointment, and the highest in the wound treated with 1% propolis and 1% nanosilver (24.0 × 1018 spin/g).

On day 10 of the treatment, the lowest concentration is seen in the wound treated with the 1% nanosilver ointment (7.8 × 1018 spin/g), and the highest in the wound treated with 1% propolis and 1% nanosilver (19.5 × 1018 spin/g).

On day 15 of the treatment, the lowest concentration was seen in the wound treated with the 1% nanosilver ointment (7.5 × 1018 spin/g), and the highest in the wounds treated with 1% propolis and 1% nanosilver (10.7 × 1018 spin/g) and the 1% propolis ointment (10.6 × 1018 spin/g).

On day 20 of the treatment, the lowest concentration is seen in the wound treated with the 1% nanosilver ointment (4.9 × 1018 spin/g), and the highest in the wound treated with 1% propolis and 1% nanosilver (9.9 × 1018 spin/g).

On day 25 of the treatment, the lowest concentration was seen in the wound treated with the 1% nanosilver ointment (1.6 × 1018 spin/g), and the highest in the wound treated with sulfathiazole (8.9 × 1018 spin/g).

## 4. Discussion

The treatment of burn wounds is a significant problem in modern medicine. An ideal topical preparation that fulfils the functions necessary for the effective treatment of thermal injuries is constantly in demand. Among the expected therapeutic properties are antimicrobial, regenerative, reparative, and anaesthetic effects.

Currently, drugs of biogenic origin, especially apitherapeutics, are of great therapeutic importance. The evaluation of the activity of the propolis ointment and silver salt of sulfathiazole described by Gregory et al. [27] showed the significant advantage of propolis in the treatment of burn wounds, compared to sulfathiazole. The course of wound treatment, formation of granulation tissue, intensification of inflammation in the tissues surrounding the wound, and quality of the scar indicate that the propolis ointment offers better results compared to the “gold standard” silver sulfathiazole. Comparable results were seen in our study.

In 2014, Adhya A. et al. [28] published a randomized study comparing the effects of treating burns with silver sulfathiazole and nanosilver. The results showed faster healing of wounds treated with 0.5% nanosilver ointment, compared to silver salt of sulfathiazole. It was observed that second-degree burns responded much better to the treatment with 0.5% nanosilver. The above observations confirm the results of our study. However, taking into account the demonstrated biotic activity of propolis and nanosilver, the experiment compared the effects of both active substances, and the synergism of nanosilver and propolis was assessed by making an ointment with a mixture of 1% propolis and 1% nanosilver. These results confirm the weak antibacterial effect of nanosilver and indicate that the active substance in the mixture is propolis.

In the international literature, there are no reports concerning the assessment of the amount of reactive oxygen species in burn wounds treated with ointments containing nanosilver and propolis. In 2013, Olczyk et al. [29] conducted an experiment to assess the amount of reactive oxygen species in burn wounds treated with 3% propolis ointment and 1% silver sulfathiazole salt. The results showed the presence of reactive oxygen species in all burn wounds. A significant decrease in the number of free radicals in the group treated with 3% propolis ointment, as compared to the silver salt of sulfathiazole, was proven. Additionally, the practical use of paramagnetic resonance in the assessment of the pharmacological activity of drugs was demonstrated.

In our study, similar results of the analysis of reactive oxygen species were observed. Additionally, it is worth noting that in wounds treated with silver salt of sulfathiazole, no downward trend in the number of free radicals was observed, from days 5 to 25 of the treatment.

The conducted study and the applied scheme of methodical procedure aimed at demonstrating the potential effectiveness of propolis and nanosilver in the treatment of burn wounds in terms of clinical practice. The study confirmed previous reports on the effectiveness of nanosilver and propolis, and their advantage over silver salt of sulfathiazole; however, no synergism of propolis and nanosilver was demonstrated [27,28,29,30,31,32,33,34].

It was confirmed that propolis, compared to the tested formulations, has a strong antimicrobial effect; significantly accelerates the healing process of burn wounds, which significantly shortens the convalescence time; has a beneficial effect on scars; and does not cause irritation or allergic reactions. The beneficial effects of the propolis ointment described in this study make it one of the most effective drugs in the treatment of wounds of various etiopathology.

## 5. Conclusions

An analysis of this study allowed to draw the following conclusions in respect to the activity and effectiveness of the ointment with 1% propolis and 1% nanosilver and sulfathiazole in the treatment of local burn wounds:The 1% propolis ointment accelerates the formation of scars, compared to the silver salt of sulfathiazole.The 1% propolis ointment has the highest antibacterial activity against G(+) bacteria.The 1% nanosilver ointment has the highest antibacterial activity against G(−) bacteria.The 1% nanosilver ointment shows the highest reduction activity of reactive oxygen species.In the group treated with the 1% propolis ointment, the process of regeneration of burn wounds was much faster compared to the other groups.There was no synergism between the action of propolis and nanosilver.

## Figures and Tables

**Figure 1 polymers-13-02312-f001:**
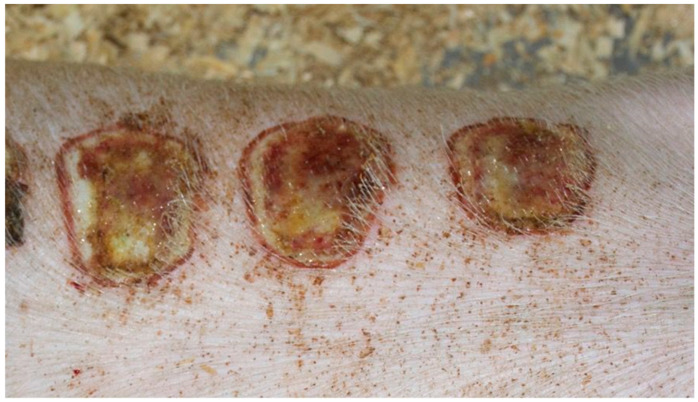
Group D1—burn wound treated with the 1% propolis ointment on day 5.

**Figure 2 polymers-13-02312-f002:**
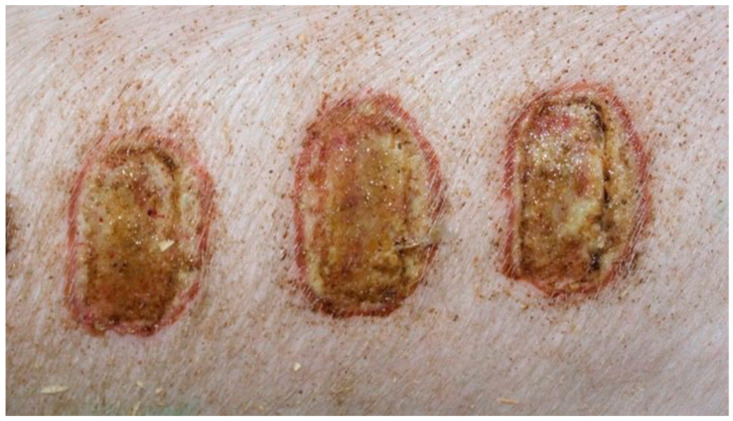
Group D2—burn wound treated with the 1% nanosilver ointment on day 5.

**Figure 3 polymers-13-02312-f003:**
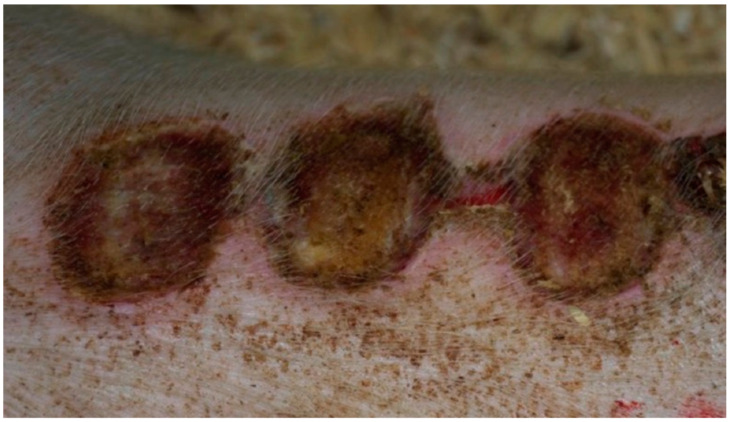
Group D3—burn wound treated with the ointment with 1% propolis and 1% nanosilver on day 5.

**Figure 4 polymers-13-02312-f004:**
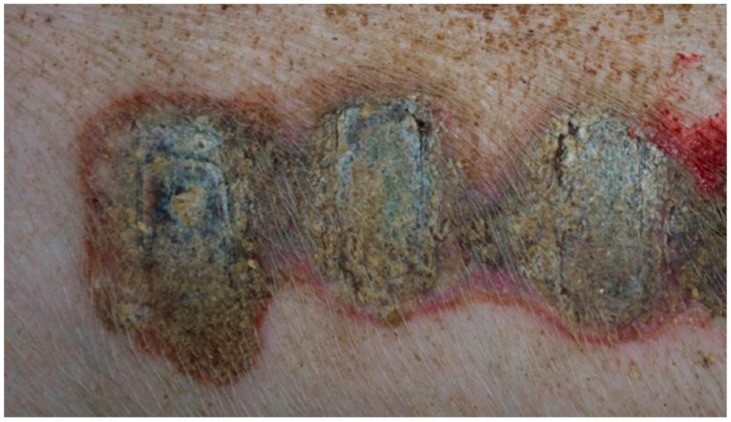
Group K—burn wound treated with sulfathiazole on day 5.

**Figure 5 polymers-13-02312-f005:**
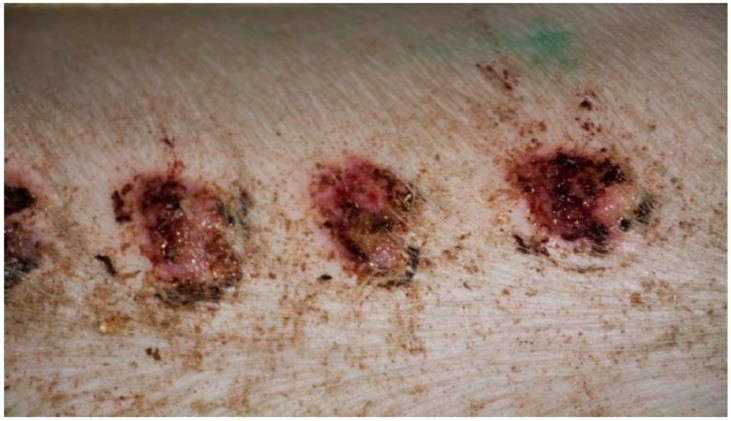
Group D1—burn wound treated with the 1% propolis ointment on day 15.

**Figure 6 polymers-13-02312-f006:**
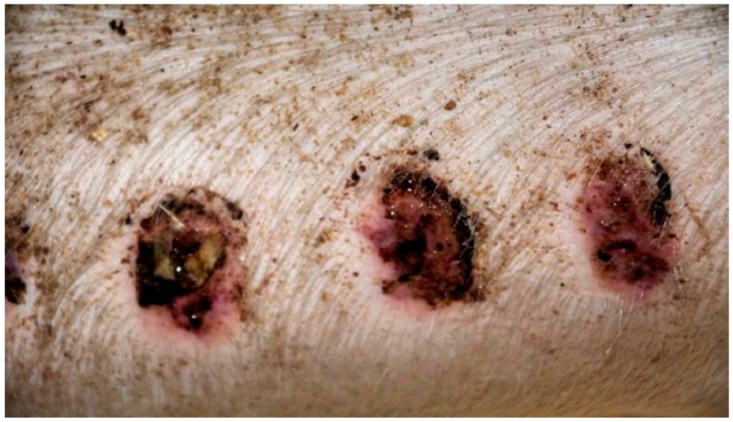
Group D2—burn wound treated with the 1% nanosilver ointment on day 15.

**Figure 7 polymers-13-02312-f007:**
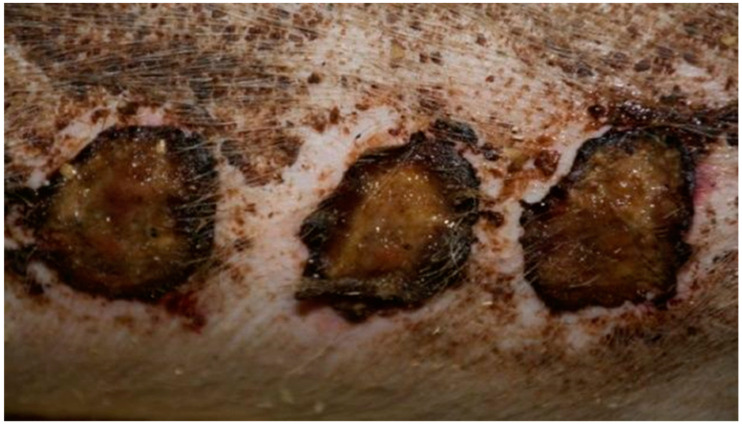
Group D3—burn wound treated with the ointment with 1% propolis and 1% nanosilver on day 15.

**Figure 8 polymers-13-02312-f008:**
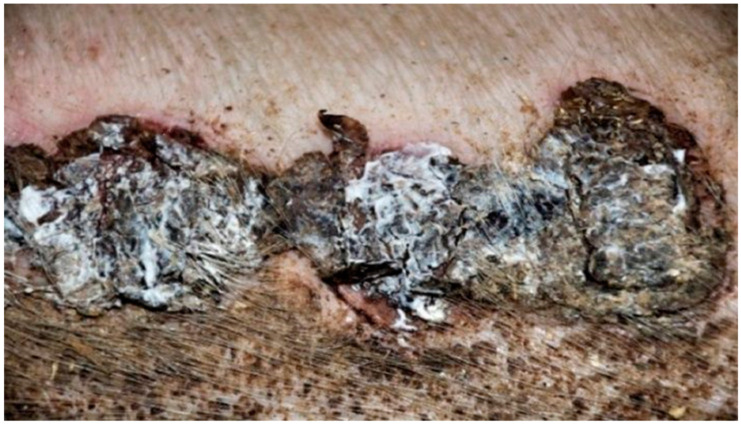
Group K—burn wound treated with sulfathiazole on day 15.

**Figure 9 polymers-13-02312-f009:**
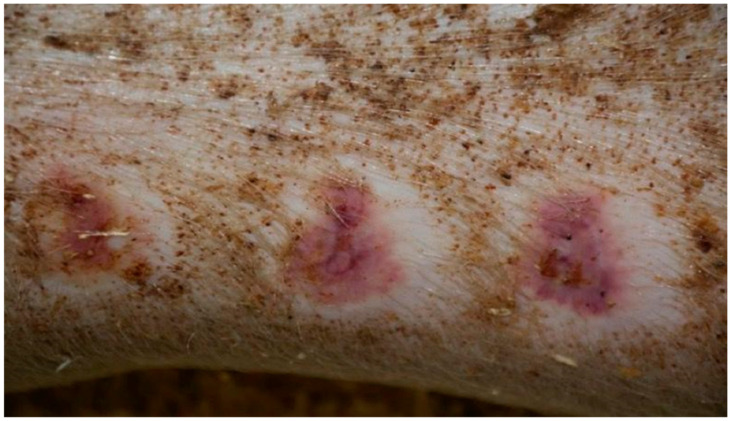
Group D1—burn wound treated with the 1% propolis ointment on day 25.

**Figure 10 polymers-13-02312-f010:**
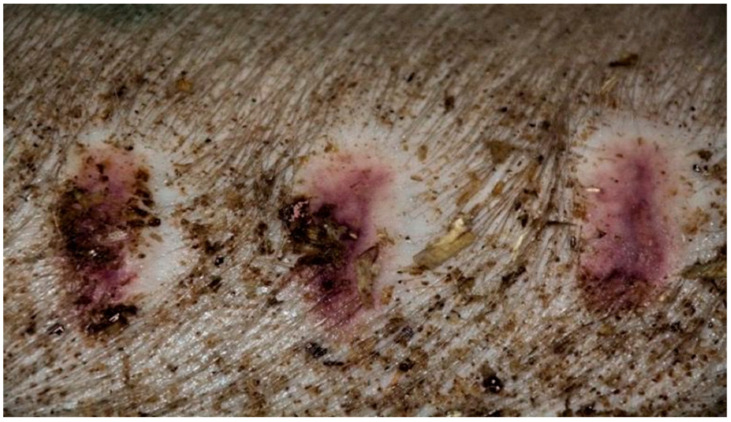
Group D2—burn wound treated with the 1% nanosilver ointment on day 25.

**Figure 11 polymers-13-02312-f011:**
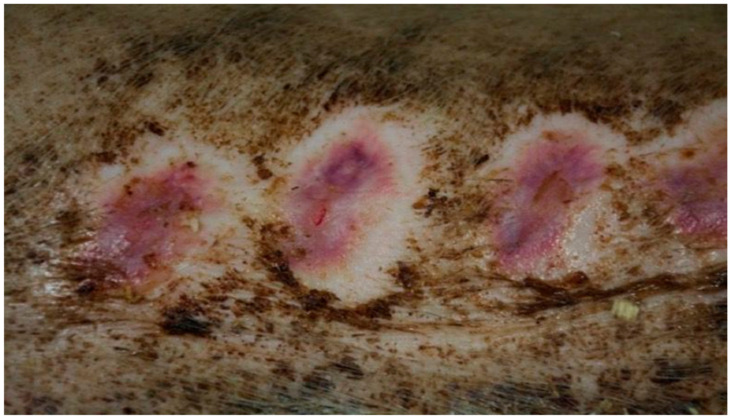
Group D3—burn wound treated with the ointment with 1% propolis and 1% nanosilver on day 25.

**Figure 12 polymers-13-02312-f012:**
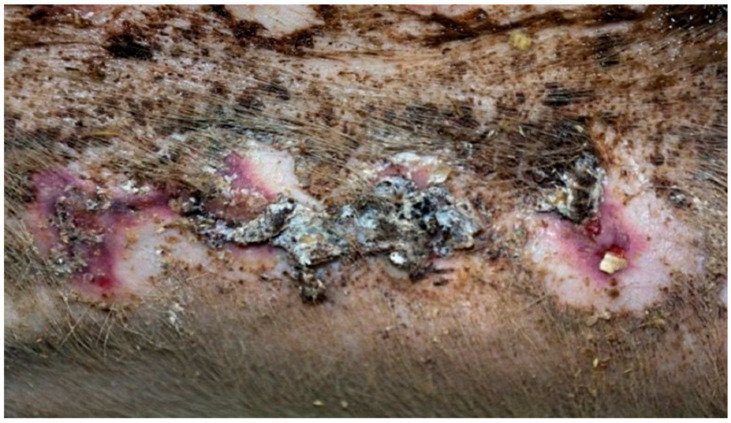
Group K—burn wound treated with sulfathiazole on day 25.

**Figure 13 polymers-13-02312-f013:**
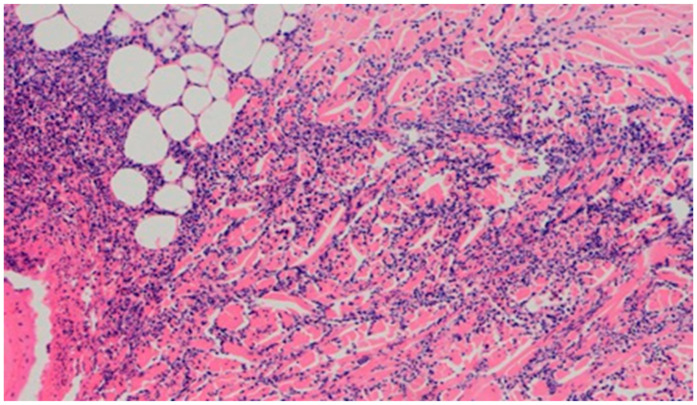
Granulocytic inflammatory infiltration on the border of the dermis and subcutaneous tissue (100×, H-E), in the wound treated with the ointment with 1% propolis and 1% nanosilver, on day 5.

**Figure 14 polymers-13-02312-f014:**
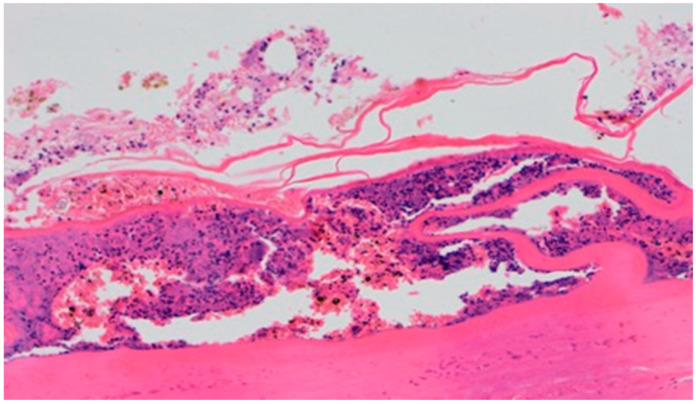
Necrosis, with granulocytic inflammatory infiltration (100×, H-E), in the wound treated with sulfathiazole, on day 5.

**Figure 15 polymers-13-02312-f015:**
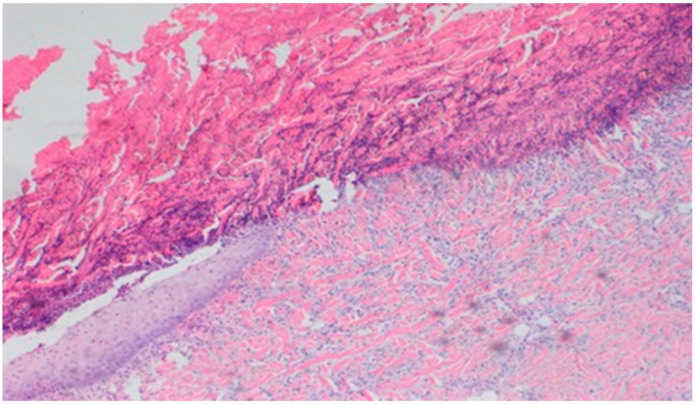
Granulocytic inflammatory infiltration, medium density.

**Figure 16 polymers-13-02312-f016:**
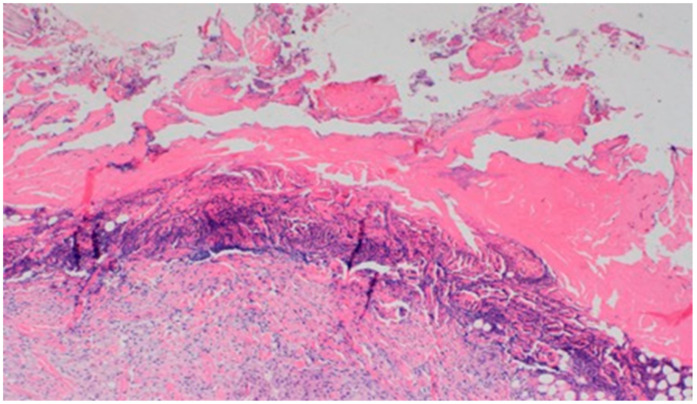
Slight superficial necrosis. Moderately intense granulocytic inflammatory infiltration in the dermis (40×, H-E), in the wound treated with the 1% nanosilver ointment, on day 10.

**Figure 17 polymers-13-02312-f017:**
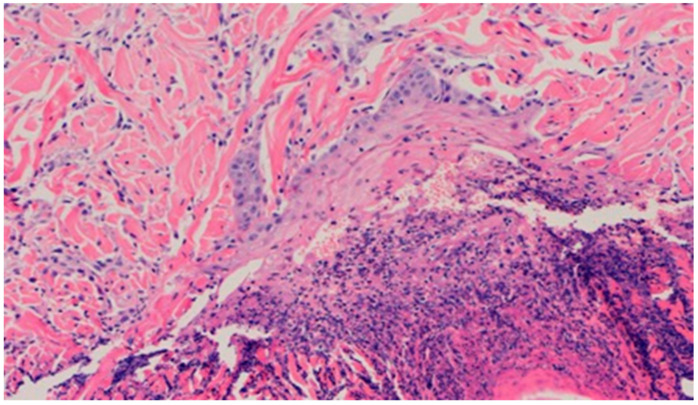
Focal epidermization under superficial necrosis (100×, H-E) in the wound treated with the 1% propolis ointment on day 10.

**Figure 18 polymers-13-02312-f018:**
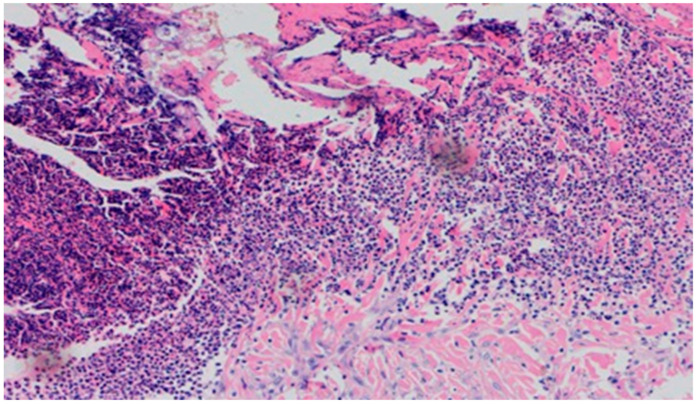
Dermis with high-density granulocytic inflammatory infiltration (100×, H-E) in the sulfathiazole-treated wound on day 15.

**Figure 19 polymers-13-02312-f019:**
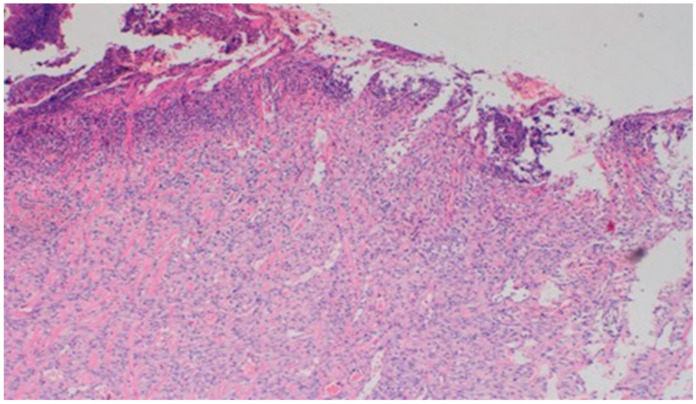
Superficial necrosis. Major neovascularization (40×, H-E) in the wound treated with the 1% nanosilver ointment on day 15.

**Figure 20 polymers-13-02312-f020:**
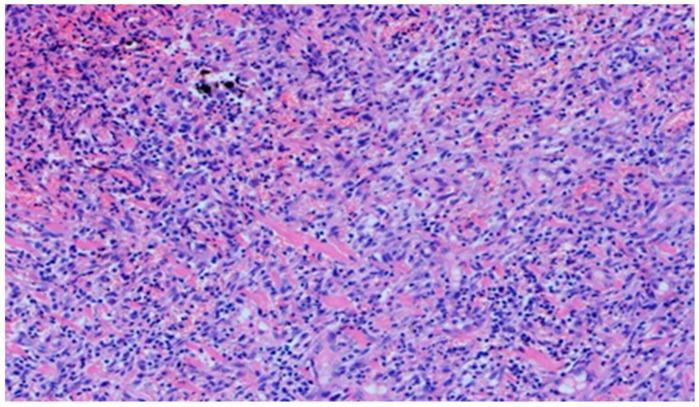
Mixed lymphocytic-granulocytic inflammatory infiltration, medium density, in the dermis (100×, H-E) in the wound treated with sulfathiazole on day 20.

**Figure 21 polymers-13-02312-f021:**
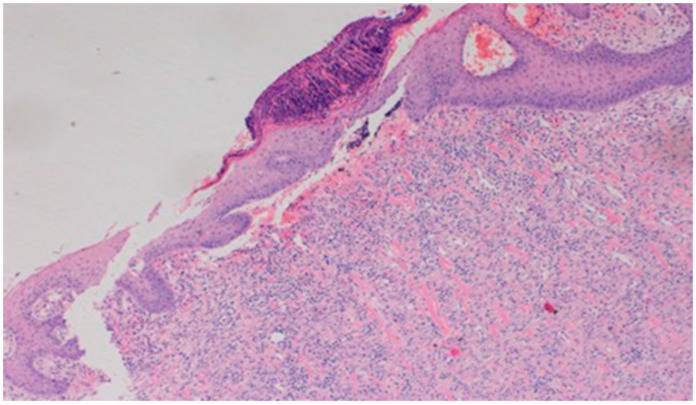
Focal, detachable necrosis on the epidermal skin lesion. Under the epithelium, a low-density lymphocytic inflammatory infiltration (40×, H-E) in the wound treated with the 1% propolis ointment on day 20 is visible.

**Figure 22 polymers-13-02312-f022:**
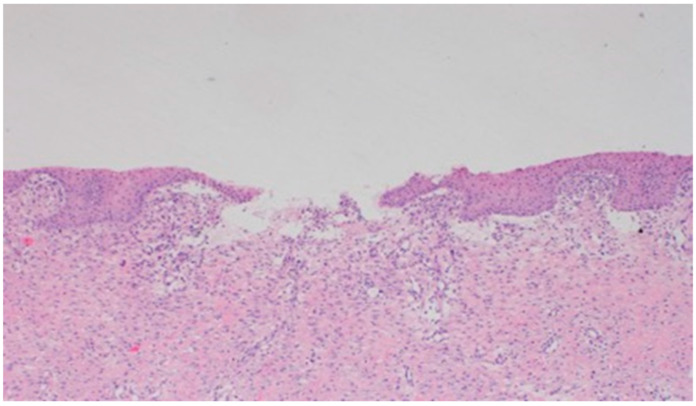
Skin wound, almost healed, without the inflammatory infiltration (40×, H-E), treated with the 1% propolis ointment, on day 25.

**Figure 23 polymers-13-02312-f023:**
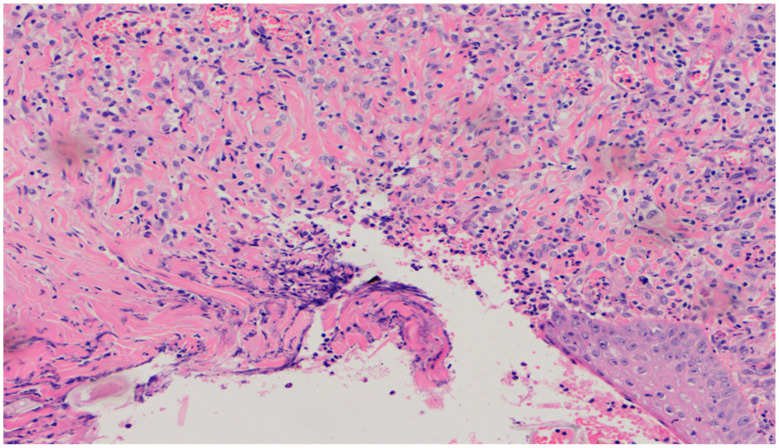
Dermis, under the ulceration with mixed (lymphocyte-predominant) inflammatory infiltration, low density, with polynuclear giant cell of foreign body type (100×, H-E), in the wound treated with sulfathiazole, on day 25.

**Figure 24 polymers-13-02312-f024:**
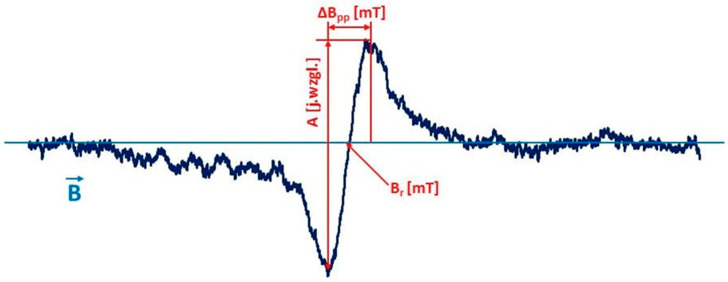
An exemplary EPR spectrum of a burnt skin sample recorded in the form of the first derivative of absorption with marked line amplitude (A), line width (ΔB_pp_), and resonance magnetic induction (B_r_).

**Figure 25 polymers-13-02312-f025:**
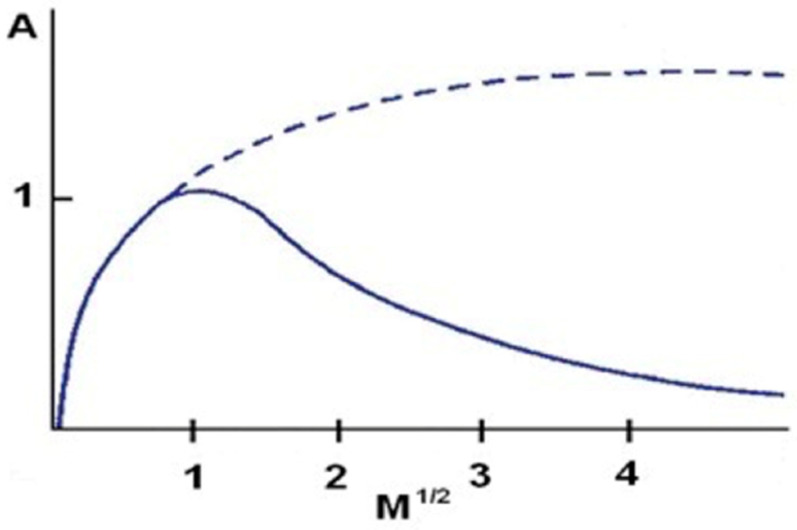
The effect of microwave power (M) on the amplitude (A) of the EPR line widened uniformly (solid line) and non-uniformly (dotted line).

**Figure 26 polymers-13-02312-f026:**
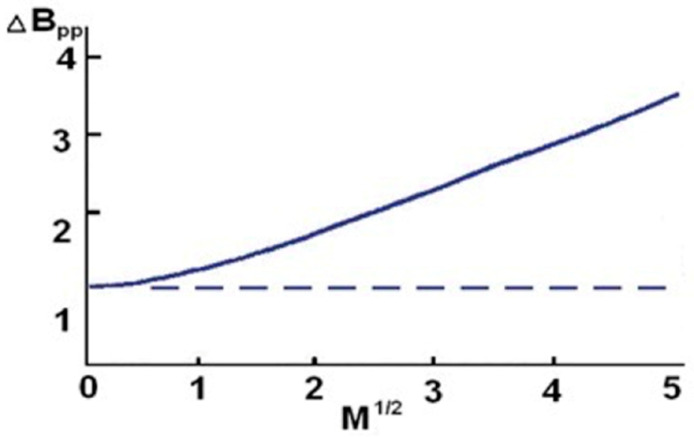
The effect of microwave power (M) on the width (ΔB_pp_) of the EPR line widened uniformly (solid line) and non-uniformly (dotted line).

**Figure 27 polymers-13-02312-f027:**
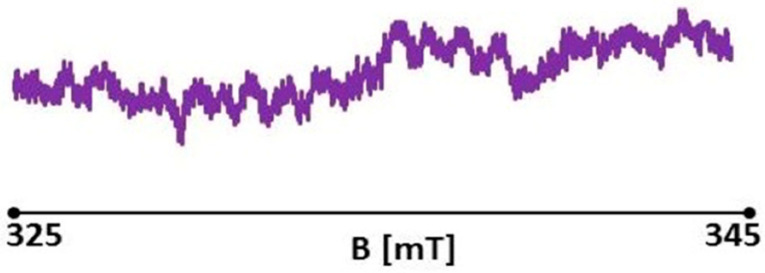
EPR spectrum of healthy skin (control). Spectrum measurement was performed with 15 dB attenuation and 2.2 mW microwave power.

**Figure 28 polymers-13-02312-f028:**
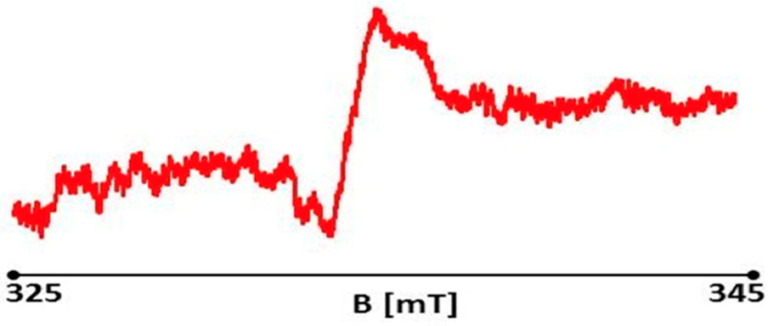
EPR spectrum of a burn wound on day 0 (control). Spectrum measurement was performed with 15 dB attenuation and 2.2 mW microwave power.

**Figure 29 polymers-13-02312-f029:**
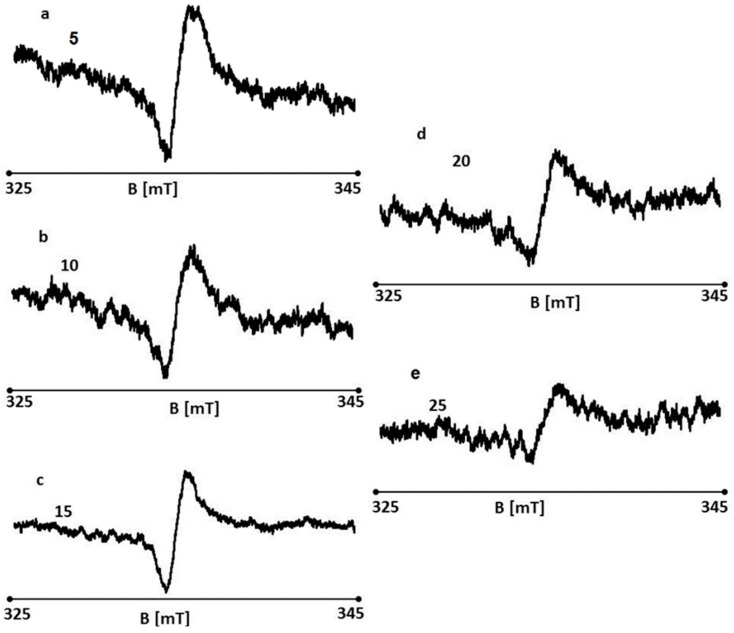
EPR spectra of burn wounds treated with the 1% propolis ointment on days 5 (**a**), 10 (**b**), 15 (**c**), 20 (**d**), and 25 (**e**) of the treatment.

**Figure 30 polymers-13-02312-f030:**
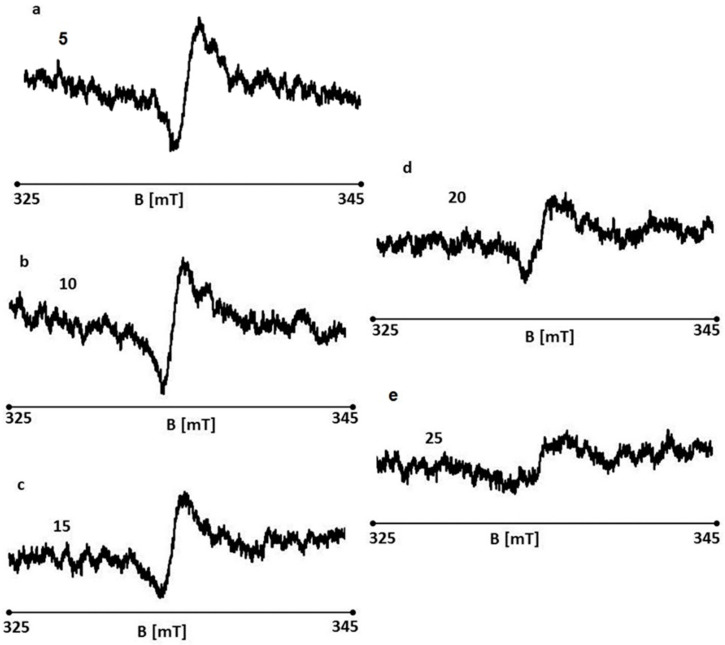
EPR spectra of burn wounds treated with the 1% nanosilver ointment on days 5 (**a**), 10 (**b**), 15 (**c**), 20 (**d**), and 25 (**e**) of the treatment.

**Figure 31 polymers-13-02312-f031:**
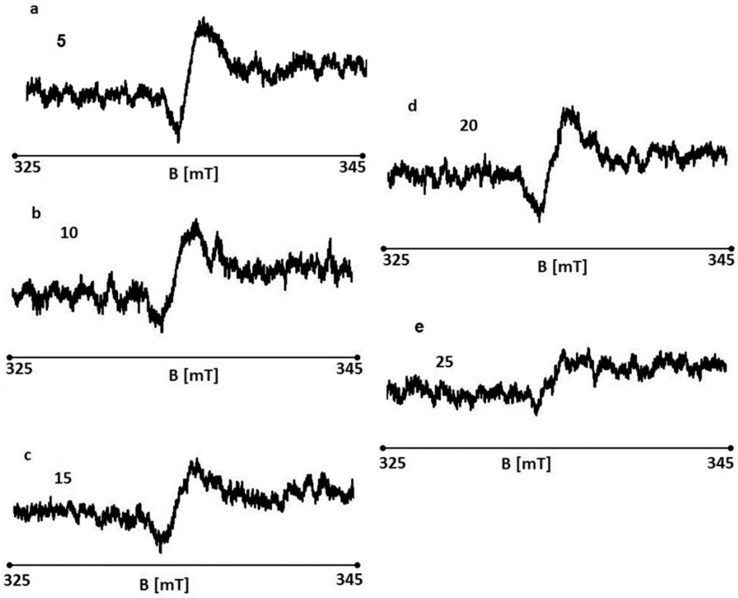
EPR spectra of burn wounds treated with the ointment with 1% propolis and 1% nanosilver on days 5 (**a**), 10 (**b**), 15 (**c**), 20 (**d**), and 25 (**e**) of the treatment.

**Figure 32 polymers-13-02312-f032:**
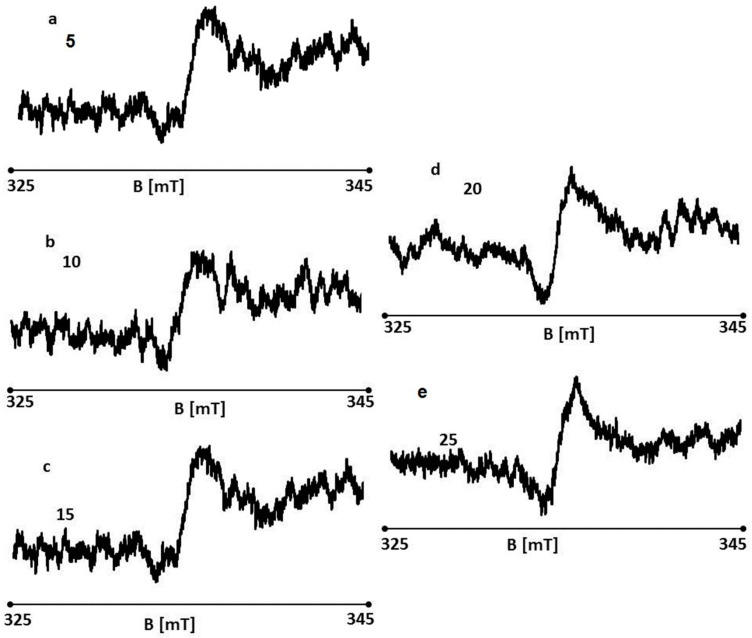
EPR spectra of sulfathiazole burn wounds on days 5 (**a**), 10 (**b**), 15 (**c**), 20 (**d**), and 25 (**e**) of the treatment.

**Figure 33 polymers-13-02312-f033:**
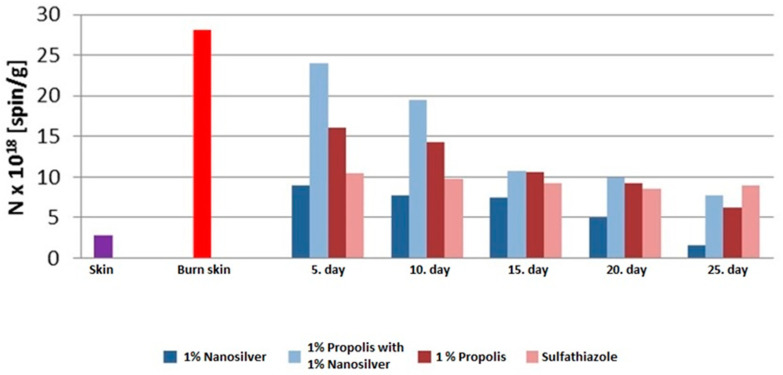
The effect of ointment therapy with 1% nanosilver (D2), 1% propolis and 1% nanosilver (D3), 1% propolis (D1), and sulfathiazole (K) on the concentration (N) of free radicals in burn wounds on days 5, 10, 15, 20, and 25 of the therapy and in healthy skin.

**Table 1 polymers-13-02312-t001:** Identification of healthy skin pathogens in pig I.

Day	*S. epidermidis*	*S. aureus*	*Streptococcus* β *haemolyticus*	*E. coli*	*Proteus mirabilis*	*Pseudomonas aeruginosa*	*Klebsiella pneumoniae*	*Candida albicans*
0	+	−	−	−	−	−	−	−
5	+++	+	−	+	−	−	−	−
10	+++	−	+	++	−	−	−	−
15	++	−	−	+	−	−	−	−
20	+++	+	+	++	−	+	−	−
25	++	+	−	++	−	++	++	−

**Table 2 polymers-13-02312-t002:** Identification of healthy skin pathogens in pig II.

Day	*S. epidermidis*	*S. aureus*	*Streptococcus* β *haemolyticus*	*E. coli*	*Proteus mirabilis*	*Pseudomonas aeruginosa*	*Klebsiella pneumoniae*	*Candida albicans*
0	+	−	−	−	−	−	−	−
5	++	+	+	+	−	−	−	−
10	+++	+	−	+	−	−	−	−
15	++	++	−	+	−	−	−	−
20	+++	−	−	+	−	++	−	−
25	−	−	−	−	−	++	−	−

**Table 3 polymers-13-02312-t003:** Identification of pathogens grown from the wound treated with the 1% propolis ointment.

Day	*S. epidermidis*	*S. aureus*	*Streptococcus* β *haemolyticus*	*E. coli*		*Proteus mirabilis*	*Pseudomonas aeruginosa*	*Klebsiella pneumoniae*	*Candida albicans*
0	+	−	−	−		−	−	−	−
5	+++	+++	+	+++		+	−	−	−
10	++	−	−	+++		++	+++	−	−
15	+++	−	−	+++	+++	−	−	−	-
20	−	+	++	+++		−	+++	+++	−
25	−	−	−	−		−	+++	+++	+

**Table 4 polymers-13-02312-t004:** Identification of pathogens grown from the wound treated with the 1% nanosilver ointment.

Day	*S. epidermidis*	*S. aureus*	*Streptococcus* β *haemolyticus*	*E. coli*		*Proteus mirabilis*	*Pseudomonas aeruginosa*	*Klebsiella pneumoniae*	*Candida albicans*
0	+	−	−	−		−	−	−	−
5	++	+	+	+		−	−	−	−
10	+++	+	−	+++		+	−	−	−
15	+++	+++	−	++	+	−	−	−	-
20	+++	+	−	+++		−	+++	++	−
25	++	+	−	−		−	+	−	−

**Table 5 polymers-13-02312-t005:** Identification of pathogens grown from the wound treated with the ointment with 1% propolis and 1% nanosilver.

Day	*S. epidermidis*	*S. aureus*	*Streptococcus* β *haemolyticus*	*E. coli*		*Proteus mirabilis*	*Pseudomonas aeruginosa*	*Klebsiella pneumoniae*	*Candida albicans*
0	+	−	−	−		−	−	−	−
5	+++	++	+	+++	+++	−	−	−	-
10	+++	−	−	+++		−	−	−	−
15	+++	+++	+++	+++		−	−	−	−
20	++	−	−	+++		+++	+++	−	−
25	−	−	−	++		+++	+++	−	−

**Table 6 polymers-13-02312-t006:** Identification of pathogens grown from the wound treated with silver salt of sulfathiazole.

Day	*S. epidermidis*	*S. aureus*	*Streptococcus* β *haemolyticus*	*E. coli*	*Proteus mirabilis*	*Pseudomonas aeruginosa*	*Klebsiella pneumoniae*	*Candida albicans*
0	+	−	−	−	−	−	−	−
5	+++	−	+++	−	−	−	−	−
10	+++	−	−	−	−	−	−	−
15	+++	+++	−	+	−	−	−	−
20	−	+++	−	+++	+++	+++	−	−
25	−	++	−	+++	+++	−	−	−

**Table 7 polymers-13-02312-t007:** Histopathological results from day 0.

Group	Inflammatory Infiltration	Type of Inflammatory Infiltration	Depth of Inflammatory Infiltration	Epidermization	Necrosis	Neovascularization
D1	Absent	Absent	Absent	Absent	Present	Absent
D2	Absent	Absent	Absent	Absent	Present	Absent
D3	Absent	Absent	Absent	Absent	Present	Absent
K	Absent	Absent	Absent	Absent	Present	Absent

**Table 8 polymers-13-02312-t008:** Histopathological results from day 5.

Group	Inflammatory Infiltration	Type of Inflammatory Infiltration	Depth of Inflammatory Infiltration	Epidermization	Necrosis	Neovascularization
D1	Average	Granulocytes	Dermis	Absent	Slight	Slight
D2	Slight	Granulocytes	Dermis	Absent	Slight	Slight
D3	Significant	Granulocytes	Adipose tissue	Absent	Slight	Slight
K	Slight	Mixed	Dermis	Absent	Significant	Slight

**Table 9 polymers-13-02312-t009:** Histopathological results from day 10.

Group	Inflammatory Infiltration	Type of Inflammatory Infiltration	Depth of Inflammatory Infiltration	Epidermization	Necrosis	Neovascularization
D1	Average	Granulocytes	Dermis	Slight	Slight	Average
D2	Average	Granulocytes	Dermis	Absent	Slight	Average
D3	Average	Granulocytes	Dermis	Absent	Slight	Average
K	Average	Granulocytes	Dermis	Slight	Slight	Average

**Table 10 polymers-13-02312-t010:** Histopathological results from day 15.

Group	Inflammatory Infiltration	Type of Inflammatory Infiltration	Depth of Inflammatory Infiltration	Epidermization	Necrosis	Neovascularization
D1	Average	Mixed	Dermis	Slight	Absent	Average
D2	Significant	Granulocytes	Dermis	Absent	Significant	Significant
D3	Average	Mixed	Dermis	Absent	Slight	Average
K	Significant	Granulocytes	Dermis	Slight	Significant	Average

**Table 11 polymers-13-02312-t011:** Histopathological results from day 20.

Group	Inflammatory Infiltration	Type of Inflammatory Infiltration	Depth of Inflammatory Infiltration	Epidermization	Necrosis	Neovascularization
D1	Slight	Lymphocytes	Dermis	Significant	Absent	Slight
D2	Average	Granulocytes	Dermis	Slight	Absent	Slight
D3	Average	Mixed	Dermis	Slight	Significant	Significant
K	Average	Mixed	Dermis	Absent	Absent	Significant

**Table 12 polymers-13-02312-t012:** Histopathological results from day 25.

Group	Inflammatory Infiltration	Type of Inflammatory Infiltration	Depth of Inflammatory Infiltration	Epidermization	Necrosis	Neovascularization
D1	Absent	Absent	Absent	Significant	Absent	Slight
D2	Absent	Absent	Absent	Significant	Absent	Slight
D3	Moderate	Lymphocytes	Dermis	Significant	Absent	Slight
K	Slight	Mixed	Dermis	Slight	Slight	Slight

## Data Availability

The data presented in this study are available on request from the corresponding author.

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
