# Peer review of "Biological Activity of Propolis Ointment with the Addition of 1% Nanosilver in the Treatment of Experimentally-Evoked Burn Wounds"

_polymers, 2021, doi:10.3390/polym13142312_

Round 1

Reviewer 1 Report

The article titled, “Biological activity of propolis ointment with the addition of 1% nanosilver in the treatment of experimentally-evoked burn wounds” is indeed an interesting study. I suggest the article to be published in Polymers journal. Few comments are below;

  1. Some stats on the accidental burnt cases and traumatic patients would be essential.
  2. The figures should be rearranged for clear analysis, now they all are scattered. The aspect ratio is high, so they can be arranged in 2 or 3 figures.
  3. The scar testing should have more detail, perhaps share some infographics of the standards, if any.
  4. Add detail of previous uses of silver nanoparticles for variant applications, 10.5923/s.textile.201401.01 and 10.1166/mat.2016.1317 could be added as references.
  5. The authors should comment on possible toxicity of silver nanoparticles too.
  6. The conclusion part should be exclusive to the main achievements of the work only. Now it has too many auxiliary sentences.

Author Response

Dear Reviewer!

  1. Statistical data has been added.
  2. List of figures added.
  3. Unfortunately, we do not have a standard. I understand this is about clinical observation? In a similar publication, wounds were treated with 0.9% NaCl - https://doi.org/10.3390/molecules181114397

5. The toxicity of nanosilver has been added.

6. Changed and abridged discussion.

If you have any questions or need further information, please feel free to ask.

Sincerely,

Jakub Staniczek

Reviewer 2 Report

This manuscript reports results on a study aimed at assessing the efficiency of an ointment containing propolis, nanosilver particles, or both, compared with a standard treatment for burns on the wound healing in 2 pigs with burns on dorsal skin.

  1. The description of the methodology to assess oxidative stress using EPR spectra needs improvement in order to understand the relationship between the spectra shown and oxidative stress, i.e. what do me mesure with EPR spectra related to oxidative stress (free radicals) and how. The parameters involved in calculations should be described, as the reader is not necessarily an expert in biophisics. References should be added.
  2. Figure 28 : please write the words in English.
  3. Figures 1-8 are not histopathological pictures but clinical pictures. Please change in the text (lines 370 and 408).
  4. Figures 13-23, hisopathological pictures : how can you make the difference between lymphocytes and granulocytes from the pictures shown ?
  5. Discussion : it could be quite shorter :
    1. The first part is redundent with the introduction or could be added to the introduction.
    2. It is not necessary to describe published studies, just indicate the differences with the present study.
    3. Focus on the novelty demonstrated in this study.
  6. Introduction, line 73 : "activation of ATPase and tetrazolium reductase" is not undestandable. Please clarify. There is no endogenous tetrazolium reductase, but endogenous reductases from Krebs cycle may reduce tetrazoilum salts used as cell viability probes.

Author Response

Dear Reviewer!

1. A more detailed description of paramagnetic resonance has been added.

2. Changed

3. Changed

4. Changed for more visible photo.

5. Changed and abridged discussion.

6. Experimenting on laboratory showed activation of glucose-6-phosphatase, NADPH2-tetrazolium reductase in rats after application of propolis. So there should be "activation of ATPase and NADPH2-tetrazolium reductase". I changed it.

If you have any questions or need further information, please feel free to ask.

Sincerely,

Jakub Staniczek

Round 2

Reviewer 1 Report

The authors have revised the manuscript accordingly.

This manuscript is a resubmission of an earlier submission. The following is a list of the peer review reports and author responses from that submission.